# Effects of Progressive Resistance Training on Cognition and IGF-1 Levels in Elder Women Who Live in Areas with High Air Pollution

**DOI:** 10.3390/ijerph17176203

**Published:** 2020-08-26

**Authors:** Edgardo Molina-Sotomayor, Humberto Castillo-Quezada, Cristian Martínez-Salazar, Marcelo González-Orb, Alexis Espinoza-Salinas, Jose Antonio Gonzalez-Jurado

**Affiliations:** 1Facultad de Artes y Educación Física, Departamento de Educación Física, Universidad Metropolitana de Ciencias de la Educación, Santiago 8320000, Chile; edgardo.molina@umce.cl (E.M.-S.); marcelo.gonzalez@umce.cl (M.G.-O.); 2Facultad de Educación y Ciencias Sociales, Carrera de Educación Física, Universidad Andrés Bello, Concepción 4600000, Chile; humberto.castillo@unab.cl; 3Facultad de Educación Ciencias Sociales y Humanidades, Departamento de Educación Física, Universidad de La Frontera, Temuco 4780000, Chile; cristian.martinez.s@ufrontera.cl; 4Escuela de Kinesiología, Universidad Santo Tomás, Santiago 8320000, Chile; alexisespinozasa@santotomas.cl; 5Physical Performance and Sports Research Centre, University of Pablo de Olavide, 41013 Sevilla, Spain

**Keywords:** exercise, physical activity, cognitive impairment, sedentarism, strength muscular environmental health

## Abstract

The aim of this study was to determine the effects of a muscular strength programme on the levels of insulin-like growth factor-1 (IGF-1) and cognitive status in elder women with mild cognitive impairment who lived in areas of high air pollution. A total of 157 women participated in the study, distributed in four groups: Active/Clean (AC *n* = 38) and Active/Pollution (AP *n* = 37), who carried out a progressive resistance training programme for 24 months, and Sedentary/Clean (SC *n* = 40) and Sedentary/Pollution (SP *n* = 42). Maximum strength in the upper and lower limbs (1RM), cognition (Mini-Mental Scale Examination (MMSE)) and blood IGF-1 were evaluated. At the beginning of the intervention, there were no differences between the groups in the assessed variables. The active groups which carried out the resistance training programme (AC and AP), registered better results in IGF-1 than the sedentary groups. These differences were statistically significant in AC vs. SC (*p* < 0.01) and AP vs. SC (*p* < 0.05). Regarding MMSE, group AC registered the highest score increases (+8.2%) (significantly better than the other groups), while group SP worsened (−7%) significantly compared to the other three groups. In conclusion, resistance training had a positive effect on IGF-1, while sedentary behaviour and air pollution had a negative effect on cognitive status.

## 1. Introduction

Regular physical activity has multiple benefits for older adults, including improved physical and cognitive health and enhanced social engagement, illustrating the value of providing exercise physiology services in the aged care sector [1], therefore, providing interventions to prevent the loss of muscle mass and improve muscular strength can help to reduce cognitive and physical limitations in the functionality of older adults [2]. The loss of muscle mass, and thus strength, has been associated with lower performance in verbal fluency tests and in all cognitive tests in adults and older adults [3]. Therefore, in the integral evaluation of elder patients it would be useful to identify all the risk factors that influence the progression of cognitive fragility and decline [4]. The decrease in muscular strength has been strongly linked to cognitive impairment [5]. Moreover, muscular strength has been demonstrated to be a better predictor of clinical results of mortality compared to muscle mass [6]. The loss of muscular power is a dominant physical determinant in the loss of functional capacity, especially when the lower limbs are affected. However, the impact of muscular strength training on cognitive function and brain structure is still under debate [7]

Furthermore, lower cognitive function has been related to lower muscular strength in older adults [8], and it has been reported that leg strength can help to predict both healthy cognitive ageing and the preservation of brain volume regardless of the genetic factors, life style and vascular risk [9]. Likewise, functional aptitude has been associated with cognitive impairment in older adults, adjusted for sociodemographic factors, chronic disease, health state, health habits and functional state [10]. Moreover, no significant training effects have been found for physical functions or brain volume in older adults after training with multicomponent exercises, which included muscular strength training [11]. However, it has been shown that resistance training increases muscular strength and improves verbal memory regardless of the load characteristics [12], finding a positive association between the change rates in grip strength and cognitive function in older adults [13]; likewise, increasing muscle strength has been reported to improve mental health in middle-aged and elderly adults [14]. In this sense, substantial changes in brain functionality have been reported, especially in the frontal lobe, accompanied by improvements in executive and neurocognitive functions and physical performance, coinciding with high levels of brain-derived neurotrophic factor (BDNF) in older women, after a muscular strength training programme [15,16]. However, further research is required on these muscular changes and their effects on older adults [17,18].

Similarly, insulin-like growth factor-1 (IGF-1) is linked to the appearance of cognitive deficits, e.g., in the tasks that evaluate executive function, attention and verbal memory [19]. Lower serum levels of IGF-1 in older adults are related to fragility [20] and dementia, which are associated with functional disability and the loss of muscular strength [21]. Some recent findings support that IGF-1 deficiency might be involved in the establishment and progression of neurodegenerative disorders [22]; likewise, IGF-1 deficiency is responsible for increased brain oxidative damage, edema and impaired learning and memory capabilities which are rescued by IGF-1 replacement therapy [23]. Therefore, some evidence indicates that interventions to mitigate IGF-1 deficiency (i.e., exercise programmes) would improve cognitive state, however, it seems that further investigation is needed to know if this association exists between IGF-1 and cognitive impairment. It is known that aerobic exercise can be a protective factor against the effects of air pollution, increasing cognition and the mechanisms of oxygen transport [24]. In a similar way, muscular strength training could be an effective stimulation to improve cognitive function in older adults. The post-exercise increase in BDNF has been reported as an acute response to a resistance training programme, whereas the level of IGF-1 remained unaltered [25]. Furthermore, nutritional supplementation in conjunction with age-appropriate resistance exercise not only boosts fat-free mass and muscle strength performance but also enhances inflammatory markers, IGF-1 levels [26] and other aspects that contribute to well-being in sarcopenic elderly [27].

Likewise, it has been suggested that air pollution can be an important risk factor for health and for the decline of the functional state [28], and it has been shown that both chronic air pollution and noise exposure may influence the physiological ageing process of the brain [29]. Some air pollutants, such as ultrafine particulate matter (PM25), have been associated with cognitive impairment, including global cognition, attention, memory and executive function, in older adults [30]. This adverse effect could vary depending on the time of exposure and the conditions of PM air pollution [31]. Furthermore, it has been suggested that exposure to air pollution can negatively affect the physical performance of older adults [32], in this sense, it has been reported an inverse association between increased air pollution (indoor and outdoor) with the decrease in manual grip muscle strength [28]. These conditions could cause cognitive deficits as a result of practicing mixed cardiovascular exercise and resistance training, compared to the basic physical activity involved in daily tasks, since approximately 1.8 times more ultrafine particles are inhaled due to the increased oxygen demand [33]. Consequently, an association between air pollution and the subsequent cognitive impairment [34] leads to a negative effect on mental health, with each increase of 1 μg/m^3^ of PM2.5 [35], causing states of neuroinflammation and neuropathology, which are associated with neurodegenerative affectations [36], as well as with poor cognitive performance in verbal and mathematics tests [37].

Table 1 shows the average triannual atmospheric contamination of the place of residence of the participants and the reference values of Chile and the World Health Organization (WHO). As can be observed, there are significant differences in the concentration of polluting particles between Viña del Mar (considered as a low-contamination area) and Pudahuel (considered as a high-contamination area).

Therefore, the aim of this study was to analyse the effects of a progressive resistance training programme on cognitive state and insulin-like growth factor-1 (IGF-1) in older women with mild cognitive impairment, comparing those who lived in areas with high air pollution with those who lived in areas with low air pollution.

## 2. Methods

### 2.1. Sample

This is an experimental, non-probabilistic, prospective, two-year longitudinal study. The participants (Figure 1) were recruited from a potential population of older women (*n* = 734). Of this potential population, 358 were patients of a health centre of the municipality of Pudahuel, in the Metropolitan Region of Santiago de Chile (Chile) (exposed to high air pollution), and 376 were patients of a health centre of the city of Viña del Mar (Chile) (exposed to low air pollution). An initial sample was obtained (*n* = 324), randomly distributed in 4 groups. The final sample (*n* = 157) was constituted by the following groups: Active/Pollution (AP, *n* = 37), age (X ± SD) 69.9 ± 5.0 years; Sedentary/Pollution (SP, *n* = 42), age 70.0 ± 6.1 years; Active/Clean (AC, *n* = 38), age 71.1 ± 4.8 years; Sedentary/Clean (SC, *n* = 40), age 69.4 ± 5.4 years (Figure 1).

All participants had an education level that allowed them to read and write perfectly without assistance. All participants were retired at the time of the intervention. They declared that they did not perform any extraordinary physical or sport activity other than the activities of daily living. The intervention period was initiated in January 2018 and was terminated in December 2019.

The inclusion criteria were the following: (1) women aged between 60 and 86 years, with up-to-date biochemical blood tests; (2) medical authorisation to do physical exercise at a moderate intensity; (3) users of the health centre of the municipality of Pudahuel (Metropolitan Region of Santiago de Chile, Chile) and senior centres of the city of Viña del Mar (Chile); (4) performed no planned physical activity in the last 5 years before the intervention; (5) lived at least 10 months per year in the last 30 years in the current place of residence; (6) completed primary education; (7) MMSE score ≤24 and ≥13 points; (8) use of non-polluting heating systems in their houses; and (9) signed the informed consent.

The exclusion criteria were the following: (1) patients with depression under pharmaceutical treatment; (2) patients with pathological conditions incompatible with physical exercise; (3) cardiovascular disease, hypopituitarism and malnutrition; (4) illiterate patients; (5) smokers; (6) lived for less than 10 months per year in the current place of residence; (7) attended less than 80% of the sessions of the training programme; (8) showed severe pathologies during the study period; and (9) incomplete measurements.

The study was approved by the Ethics Committee of University of Chile, Santiago de Chile. The process followed the guidelines of the Declaration of Helsinki, approved by the World Medical Association [38].

### 2.2. Procedures and Temporalisation.

Cognitive impairment was evaluated by the Mini-Mental State Examination (MMSE), one of the most widely used and studied instruments worldwide. Either used alone or combined with more comprehensive instruments, this test allows assessing cognitive function and screening of dementia conditions. The MMSE covers six cognitive dimensions: time orientation, space orientation, registration, attention-calculation, recall memory and language [39]. The Spanish version has been previously validated in a sample of Chilean older adults [40], and it has been used in populations of Hispanic elders [41]. In this study, a total score of 30 points was used, with a cut-off point of ≤24 for the diagnosis of or mild cognitive impairment (MCI) [42,43,44]. The use of prediction equations to determine a 1RM in older adults is a practical tool for strength training and to encourage participation in resistance training in this population. Prescribed programmes using estimated 1RM values allow calculating the appropriate training load intensity [45]. Recent studies have shown their reliability and validity to estimate 1RM [46]. The equations developed by [47] were applied in the present study.

The exercise used to test the maximum strength of the lower limbs was “45º Leg Press”. The strength of the upper limbs was evaluated through “90º Seated Chest Press”. The resistance training exercises performed during the intervention programme are shown in Figure 2.

The 1RM estimations were conducted following these phases (Figure 3):

Phase 1: Week -1. In the week prior to the initial estimation of 1RM (pre-test), two sessions of technical and diagnostic adaptation were held, which consisted of carrying out submaximal strength exercises on two non-consecutive days, with 48 h between them. On the first day, the participants performed 2 series of 6–8 repetitions in “90º Seated Chest Press” and 5–7 repetitions in “45º Leg Press”. The intensity was estimated from the OMNI scale [48], thus it was 30%–50% of the estimated 1RM. The participants rested for 3–4 min between series.

Phase 2: Week 0. Pre-test. In the following week, the initial tests for the estimation of the theoretical 1RM were carried out. The sessions began with a general warm-up based on general joint mobility (5′) and a specific warm-up, which consisted of 3 series of 10, 12 and 15 repetitions with very light loads (20–30% 1RM, OMNI scale), in each of the two exercises used for the muscular strength tests. Then, the participants conducted two attempts with the same submaximal loads to failure, with 5 min of rest between the two attempts. The loads used were based on the estimations recorded in Phase 1 (technical and diagnostic adaptation). The series considered to estimate the theoretical 1RM were those in which the repetitions were 8–10 RM, taking the largest number of repetitions.

Phase 3. Weeks 1–101 (Months 1–24). Period of intervention based on muscular strength training against resistance. During this period, the protocol of Phase 2 was followed 5 times (Table 2).

Phase 4. Week 102 (Month 25). Post-test. The protocol of Phase 2 was repeated.

Table 2 shows the characteristics of the progressive resistance training programme. This programme consisted of multi-joint exercises that were performed twice per week in 90-min sessions. Each session began with a 15-min warm-up of vegetative activation through low-intensity aerobic exercises and joint mobility exercises for the shoulders, spine, hips and knees. At the end of each session, the cooldown consisted of assisted static stretching exercises and joint amplitude of the main muscle groups and articular chains, for approximately 10 min. Every 4 months, the theoretical 1RM estimation tests were repeated, in order to adjust the training stimuli to the individual adaptations and guarantee the precision of the intensity of the training load applied throughout the entire intervention period.

During the intervention period of two years, 7.2% of the training sessions in the high-pollution area were cancelled after a preventive environmental alert was declared, because of the high average concentrations of particulate matter (PM10) reaching above 200 µg/m3 in 24 h [49].

Forearm blood samples were taken to determine IGF-1. This was conducted from 8:00 to 10:30 a.m., extracting 4–5 mL of blood after fasting for 10–12 h. The concentration levels of IGF-1 were obtained by chemiluminescence using an IDS-iSYS^®^ analyser (IDS Ltd., Boldon, UK)), calibrated to the reference pattern WHO NIBSC IS 02/254, with a minimum measuring concentration of 8.8 ng/mL.

### 2.3. Statistical Analysis

IBM SPSS 23^®^ software (SPSS Inc. Chicago, IL, USA) was used for the statistical analysis. The descriptive statistic, mean and standard error were calculated. To estimate the reliability of averages, a 95% confidence interval was calculated. Regarding the intragroup pre-post differences, paired Student t-test or Wilcoxon test was conducted depending on the Shapiro–Wilk normality test. A multivariate general linear model (group factor) was applied for intergroup hypothesis testing. To analyse the pairwise differences, the Bonferroni or Games–Howell post hoc test (depending on the Levene test) were carried out. The effect size was measured through partial eta squared. Statistical significance was established at *p* < 0.05.

## 3. Results

Table 3 shows the intergroup comparisons of the results obtained at the beginning of the experiment (pre-test). The mean values of the four groups at the beginning of the experiment indicated that there were no significant differences between them (*p* > 0.05). Therefore, the post hoc pairwise comparisons (Bonferroni adjustment) do not show significant differences.

Table 4 shows the intergroup comparisons of the results obtained at the end of the experiment (post-test). Significant differences were found between the groups in all the analysed variables, except for IGF-1. The post hoc analysis shows the differences detected in the pairwise comparisons. The same superscripts (a, b, c, d, e, f) in mean values indicate significant differences between groups.

The AC group (active and in an environment of clean air) obtained the best results in all variables. The significant differences in muscular strength between active and sedentary groups were expected because of the training programme. Regarding the MMSE, the AC group showed a better global score in the post-tests than the rest of the groups, whereas the SP group (sedentary and in an environment of polluted air) obtained the worst score, with statistically significant differences.

Figure 4 shows the results of the intragroup comparison before and after the intervention period.

The variables of muscular strength in the participants who carried out the training programme improved in both muscle groups after the intervention period. Such improvements were greater in AC than in AP (active and in an environment of polluted air), although they were statistically significant in both cases. The SP group showed a significant decrease in muscular strength, in both the lower and upper limbs.

With respect to IGF-1, the results obtained by the four groups were similar to those obtained in muscular strength. That is, the two active groups (AC and AP) showed significantly higher blood levels of this protein, whereas the SP group showed a significant decrease in its concentration.

Regarding cognitive impairment analysed through the MMSE, it was observed that all groups improved their scores significantly, except for the sedentary group who lived in areas of high air pollution, who showed a significant decrease.

Figure 5 compares the percentage change between the post-tests and pre-tests among the four groups. As was expected, the active groups improved the levels of muscular strength to a greater extent.

The active groups who carried out the training programme obtained better results, not only in muscular strength. As can be observed between the sedentary groups, the group that lived in an environment of polluted air showed worse results than the group who lived in an area of clean air.

## 4. Discussion

The aim of this study was to determine the effects of a muscular strength training programme on cognitive state and IGF-1 levels in older women with mild cognitive impairment, as a function of the level of air pollution.

### 4.1. Muscular Strength

The scientific literature shows that resistance training is a powerful intervention to counteract the loss of muscular strength and muscle mass caused by the lack of muscle use, which leads to situations of physiological vulnerability and its weakening effects on physical functioning, mobility, autonomy, chronic diseases and, thus, the quality of life and healthy life expectancy, especially in older adults [50].

One of the main findings of the results of this study is that a population of older adults subjected to a controlled muscular strength training programme with individualised loads can significantly improve maximum muscular strength, as was observed in the AC and AP groups (*p* < 0.001 and *p* < 0.01, respectively) (Figure 4). These results were expected and are in line with those of previous studies regarding resistance training in older adults [51,52].

Similarly, the results obtained in the intragroup comparisons show that a sedentary lifestyle in older women who live in areas with high air pollution leads to the worsening of muscular strength levels, with very significant decreases (SP group: *p* < 0.001) (Figure 4). It is accepted that the ageing process, even in the absence of chronic disease, is associated with biological changes that can contribute to decreases in skeletal muscle mass, strength and function [50,53].

The intergroup comparisons showed that the SP group obtained worse results than the rest of the participants of the study. Not only did it not improve, but it also showed a muscular strength decrease of 9.9% and 5.3% (Figure 5), with statistically significant differences when compared to the active groups. A study carried out in the Netherlands demonstrated that exposure to air pollution may adversely affect the physical performance of older adults [32]. On the other hand, resistance training and clean air (AC group) showed the best results in muscular strength, with statistically significant differences with respect to the sedentary groups (SC and SP), and it also obtained better results than the AP group (who carried out the same training programme), especially in upper limb strength, where the differences were statistically significant (*p* < 0.5). These results are in line with those of studies which demonstrate that physical performance and adaptations to training stimuli are lower in environments of high air pollution [24,54], specifically in muscular strength, hence a recent study suggested that air pollution might be an important risk factor of poorer health and functional status as indicated by hand-grip strength [28].

### 4.2. Insulin-Like Growth Factor-1

Regarding IGF-1, the results show that resistance training is a determinant factor in the changes observed in the different study groups.

The intragroup comparisons show that the plasma levels of IGF-1 clearly improved in the two active groups during the intervention (AC: *p* < 0.001 and AP: *p* < 0.05), whereas it decreased in the two sedentary groups. Specifically, in the SP group, it decreased significantly (*p* < 0.01) (Figure 4). This is in line with other studies which have associated low levels of IGF-1 with older adults [55], as well as with low levels of physical condition [56].

The intergroup comparisons also show the effect of training on IGF-1, with an opposite trend in the active groups, who increased their concentrations of IGF-1 (AC: 5.3% and AP: 4%), whereas the sedentary groups obtained a decrease in this protein (SC: −0.7% and SP: −2.9%) (Figure 5). The SP group obtained especially negative results, showing a significantly different decrease with respect to the active groups (AC vs. SP: *p* < 0.001 and AP vs. SP: *p* < 0.05). It has been reported that basal levels of IGF-1 did not change systematically due to resistance training, nor did they differ between young men and older men [57], however these results are in line with those reported in a recent study in which untrained older women were subjected to a strength training programme, improving their IGF-1 levels by 7–10% in twelve intervention weeks, whereas the control group decreased by 2.2% [58]. Likewise, a group of older adults with sarcopenic obesity who engaged in a resistance training programme demonstrated increased muscle strength performance and serum IGF-1 level [26].

### 4.3. Cognitive State (Mini-Mental State Examination)

The most relevant findings obtained in this study are probably those related to cognition. All participants initiated the study with a diagnosis of mild cognitive impairment (MMSE score <24). The initial situation was similar in all four groups at the beginning of the experiment (Table 3).

The intragroup comparisons (Figure 4) show completely different behaviours in the AC and SP groups. While the AC group improved significantly (*p* < 0.001), the SP group worsened its cognitive state significantly (*p* < 0.001). The combination of a sedentary lifestyle and living in an area with high air pollution seems to have a considerable negative effect on the cognitive state. These results are similar to those reported in different studies, which conclude that sedentary behaviours could be associated with worse cognition levels [59,60,61]. On the other hand, the combination of living in an environment with low levels of contamination and having training habits improves muscular strength and has clear positive effects on the cognitive state. In the specific case of the AC group, the initial situation of mild cognitive impairment was practically reverted, obtaining 23.9 points in the MMSE score at the end of the intervention (Table 4). In this sense, the intergroup comparisons (Figure 5) show that the combination of clean air and resistance training (AC group) caused greater improvements in the cognitive state with respect to the other groups, obtaining statistically significant differences in the three pairwise comparisons. These results are in line with those of other studies, which report that different strength training programmes (resistance training) improve cognitive function. It seems to be clearly demonstrated that controlled and planned physical exercise improves cognitive impairment, besides physical and psychosocial health in older adults [1,14,62,63,64].

Similarly, although the active groups improved their MMSE score, the AC group improved this parameter further than the AP group (*p* < 0.01). The two groups carried out the same training programme; however, it seems that the “clean air vs. polluted air” factor influenced the differences in the results. In this sense, statistically significant differences were obtained between the two sedentary groups (SC vs. SP, *p* < 0.001). It has been demonstrated that the level of air pollution has a negative effect on the cognitive processes, especially in older adults [30,65,66].

A weakness of the study is the low training frequency, as two training sessions a week perhaps is a training stimulus not too strong to induce larger significant effects. Another weakness is the rate of participant dropouts that was 51,5% of those who started the intervention period, an important percent, although it is very complicated to keep adherence and motivation for 24 months in this kind of investigation.

## 5. Conclusions

The obtained results show that a progressive resistance strength programme, at low-moderate intensity, individualised and controlled, caused significant improvements in maximum muscular strength in older women. Likewise, muscular strength training increased the blood levels of IGF-1, although this parameter was not significantly influenced by the “clean air vs. polluted air” factor. Similarly, the groups that carried out a progressive resistance strength programme improved their cognition levels compared to the groups who kept a sedentary lifestyle. Furthermore, the level of air pollution did have a negative influence on cognitive impairment, thus the groups who lived in areas of high air pollution obtained significant differences in the MMSE scores with respect to those who lived in areas of low air pollution.

Numerous authors agree on the need for further longitudinal studies to determine with more precision the independent effect of different sedentary behaviours on cognitive functions [67,68,69], and to establish the most efficient type of exercise and characteristics of training programmes to counteract cognitive impairment [59,64,70].

## Figures and Tables

**Figure 1 ijerph-17-06203-f001:**
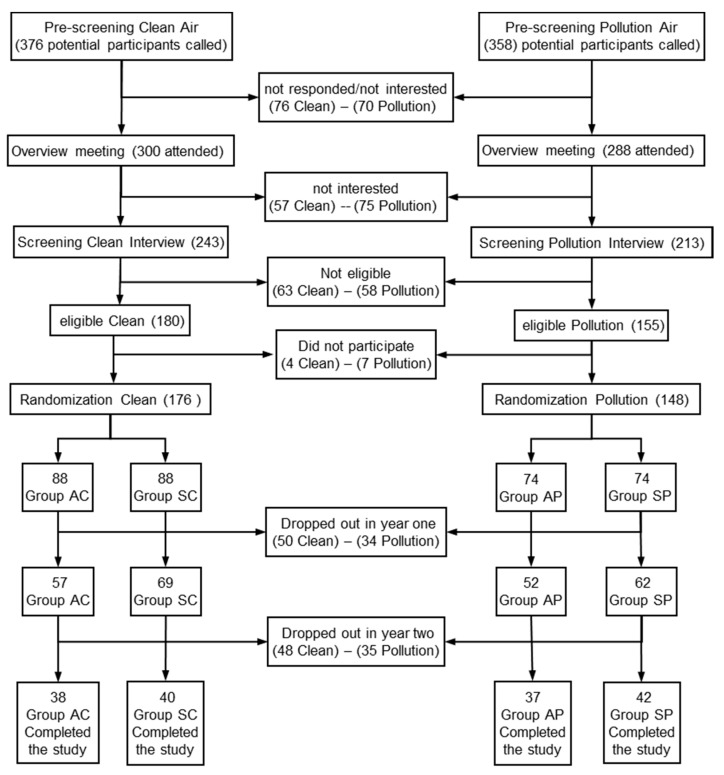
Sequence for recruitment screening. Group AC (Physical Activity and Clean Air). Group AP (Physical Activity and Polluted Air). Group SC (Sedentary and Clean Air). Group SP (Sedentary and Polluted Air).

**Figure 2 ijerph-17-06203-f002:**
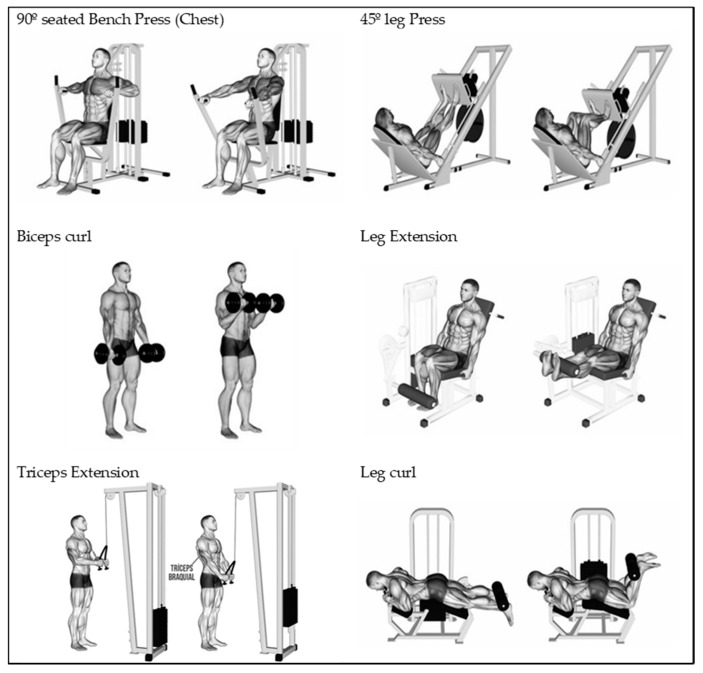
Training and testing exercises.

**Figure 3 ijerph-17-06203-f003:**
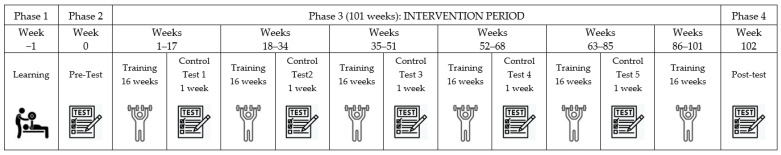
Temporalisation.

**Figure 4 ijerph-17-06203-f004:**
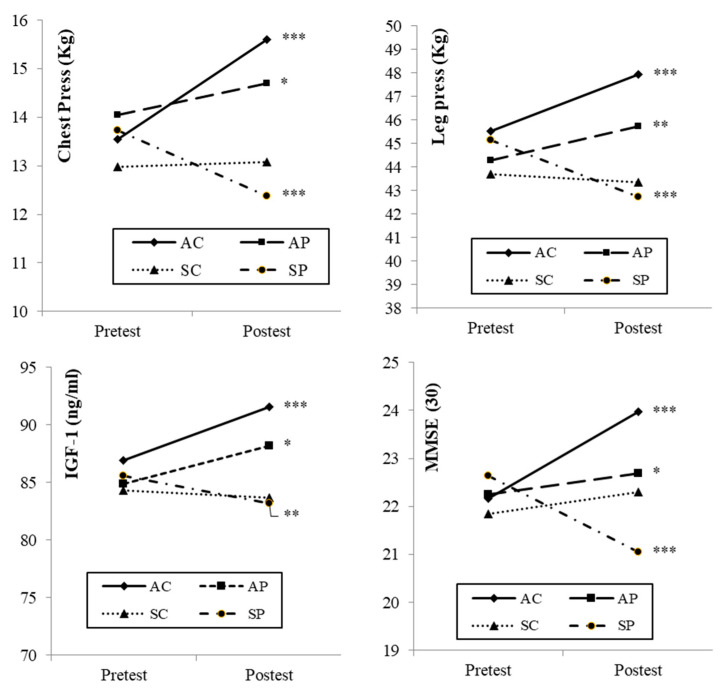
Intragroup post-test–pre-test comparisons. Wilcoxon’s or Student’s *t*-test according to normality. Significant differences (* *p* < 0.05; ** *p* < 0.01; *** *p* < 0.001). AC: Active Clean; AP: Active Pollution; SC: Sedentary Clean; SP: Sedentary Pollution.

**Figure 5 ijerph-17-06203-f005:**
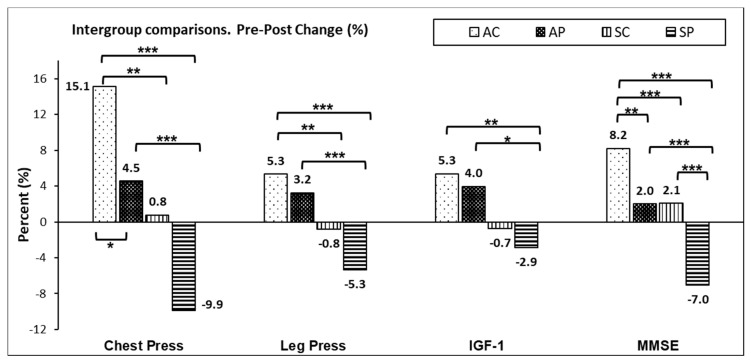
Intragroup comparisons changes (post-test–pre-test differences). GLM multivariate analysis. One factor. Post hoc pairwise comparisons (Bonferroni or Games–Howell according to Levene test): (* *p* < 0.05; ** *p* < 0.01; *** *p* < 0.001). AC: Active Clean; AP: Active Pollution; SC: Sedentary Clean; SP: Sedentary Pollution.

**Table 1 ijerph-17-06203-t001:** Record of the average triannual atmospheric concentrations (2017–2018–2019) and reference limit values of Chile and the WHO.

Area	PM_10_ µg/m^3^	PM_2.5_ µg/m^3^	NO_2_ µg/m^3^	O_3_ µg/m^3^	SO_2_ µg/m^3^
Pudahuel	67.5	29.2	38.7	26.8	NI
Viña del Mar	34.3	12.8	NI	NI	NI
Chile	50 **	20 **	100 **	120 *	60 **
WHO	20 **	10 **	40 **	100 *	20/24 h

PM10 (articulate matter diameter ≤10 µm); PM2.5 (particulate matter diameter ≤2.5 µm); NO2 (nitrogen dioxide); O_3_ (ozone); SO_2_ (sulphur dioxide); NI (No Information). * Percentile (P99) of the daily maximum 8 h. ** Annual average; source: National Air Quality Information System of Chile (SINCA).

**Table 2 ijerph-17-06203-t002:** Timetable of the resistance training programme.

Weeks	0	1–17		18–34		35–51		52–68		63–85		86–101	102
Intensity		1RM 30–40%		1RM 35–40%		1 RM 40–45%		1RM 40–45%		1RM 45–50%		1RM 50–55%	
**Bench** **press**	Pre-test ^¥^	1–2 set; 6–8 rep 30–60 s *	Control test 1 ^¥^	2–3 set; 6–8 rep 30–60 s *	Control test 2 ^¥^	2–4 set; 5–7 rep 1–2 min *	Control test 3 ^¥^	3–4 set; 4–6 rep 2–3 min *	Control test 4 ^¥^	2–3 set; 3–5 rep 3–4 min *	Control test 5 ^¥^	2–3 set; 2–4 rep 3–4 min *	Post-test ^¥^
**Biceps** **curl**		1–2 set; 6–8 rep 30–60 s *		2–3 set; 6–8 rep 30–60 s *		2–4 set; 5–7 rep 1–2 min *		3–4 set; 4–6 rep 2–3 min *		2–3 set; 3–5 rep 3–4 min *		2–3 set; 2–4 rep 3–4 min *	
**Triceps** **extension**		1–2 set; 6–8 rep 30–60 s *		2–3 set; 6–8 rep 30–60 s *		2–4 set; 4–6 rep 1–2 min *		3–4 set; 4–6 rep 2–3 min *		2–3 set; 3–5 rep 3–4 min *		2–3 set; 2–4 rep 3–4 min *	
**Leg** **press**		1–2 set; 5–7 rep 30–60 s *		2–3 set; 5–7 rep 30–60 s *		2–4 set; 4–6 rep 1–2 min *		3–4 set; 3–5 rep 2–3 min *		2–3 set; 2–4 rep 3–4 min *		2–3 set; 2–3 rep 3–4 min *	
**Leg** **extension**		1–2 set; 5–7 rep 30–60 s *		2–3 set; 5–7 rep 30–60 s *		2–4 set; 4–6 rep 1–2 min *		3–4 set; 3–5 rep 2–3 min *		2–3 set; 2–4 rep 3–4 min *		2–3 set; 2–3 rep 3–4 min *	
**Leg** **curl**		1–2 set; 4–6 rep 30–60 s *		2–3 set; 4–5 rep 30–60 s *		2–4 set; 3–4 rep 1–2 min *		3–4 set; 2–4 rep 2–3 min *		2–3 set; 2–3 rep 3–4 min *		2–3 set; 1–2 rep 3–4 min *	

* Rest time between sets. ^¥^ 90° Seated Chest Press and 45° Leg Press.

**Table 3 ijerph-17-06203-t003:** Intergroup comparisons: PRE-test *.

Variables	Active Clean (*n* = 38)		Active Pollution (*n* = 37)		Sedentary Clean (*n* = 40)		Sedentary Pollution (*n* = 42)		Intergroup Comparisons ^§^		
	M ± SE	CI (95%)	M ± SE	CI (95%)	M ± SE	CI (95%)	M ± SE	CI (95%)	F	*p* Value	PartialEta^2^
**Chest Press (Kg)**	13.5 ± 0.35	(12.8–14.2)	14.1 ± 0.36	(13.3–14.8)	12.9 ± 0.34	(12.3–13.7)	13.7 ± 0.34	(13.1–14.4)	1.68	0.174	0.032
**Leg Press (Kg)**	45.5 ± 0.68	(44.2–46.9)	44.3 ± 0.70	(42.9–45.7)	43.7 ± 0.66	(42.4–45.1)	45.1 ± 0.64	(43.9–46.4)	1.53	0.210	0.029
**IGF-1 (ng/mL)**	86.9 ± 3.89	(79.2–94.6)	84.7 ± 4.00	(76.9–92.8)	84.3 ± 3.80	(76.8–91.8)	85.6 ± 3.70	(78.3–92.9)	0.084	0.968	0.002
**MMSE**	22.2 ± 0.3	(21.6–22.7)	22.2 ± 0.31	(21.6–22.9)	21.8 ± 0.29	(21.1–22.7)	22.6 ±0.29	(22.1–23.2)	1.87	0.136	0.036

* Mean ± standard error (M±SE) and 95% confidence interval (CI 95%). Effect size by partial Eta^2^ (0.01: small, 0.06: medium, 0.14: large); ^§^ GLM multivariate analysis. One factor (group). MMSE: Mini-Mental State Examination (maximum score 30). IGF-1: insulin-like growth factor 1.

**Table 4 ijerph-17-06203-t004:** Intergroup comparisons: POST-test *.

Variables	Active Clean (*n* = 38)		Active Pollution (*n* = 37)		Sedentary Clean (*n* = 40)		Sedentary Pollution (*n* = 42)		Intergroup Comparisons ^§^		
	M ± SE	CI (95%)	M ± SE	CI (95%)	M ± SE	CI (95%)	M ± SE	CI (95%)	F	*p* Value	PartialEta^2^
**Chest Press (Kg)**	15.6 ± 0.43 ^a,b^	(14.8–16.4)	14.7 ± 0.44 ^c,d^	(13.8–15.6)	13.1 ± 0.41 ^a,c^	(12.2–13.9)	12.4 ± 0.41 ^b,d^	(11.6–13.2)	12.42	0.000	0.197
**Leg Press (Kg)**	47.9 ± 0.73 ^a,b^	(46.5–49.4)	45.7 ± 0.75 ^c^	(44.2–47.2)	43.3 ± 0.71 ^a^	(41.9–44.7)	42.7 ± 0.69 ^b,c^	(41.4–44.1)	11.11	0.000	0.180
**IGF-1 (ng/mL)**	91.6 ± 3.7	(69.8–113.3)	88.2 ± 3.8	(65.9–110.6)	83.7 ± 3.6	(76.5–90.9)	83.2 ± 3.5	(62.5–103.8)	1.173	0.322	0.023
**MMSE**	23.9 ± 0.29 ^a,b,c^	(23.4–24.5)	22.7 ± 0.3 ^a,d^	(22.1–23.3)	22.3 ± 0.28 ^b,e^	(21.7–22.8)	21.1 ± 0.27 ^c,d,e^	(20.5–21.6)	18.39	0.000	0.266

* Means ± standard error (M ± SE) and 95% confidence interval (CI 95%). Effect size by partial Eta^2^ (0.01: small, 0.06: medium, 0.14: large); ^§^ GLM multivariate analysis. One factor (group). Post hoc pair wise Bonferroni or Games–Howell comparisons according to Levene test (same superscripts indicate significant differences). MMSE: Mini-Mental State Examination (maximum score 30). IGF-1: insulin-like growth factor 1. The significant *p*-values are in bold.

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
