# Peer review of "Effects of Progressive Resistance Training on Cognition and IGF-1 Levels in Elder Women Who Live in Areas with High Air Pollution"

_ijerph, 2020, doi:10.3390/ijerph17176203_

Round 1

Reviewer 1 Report

ijerph-868965-peer-review-v1

The study wants to explore “Effects of progressive resistance training on cognition and IGF-1 levels in elder women who live in areas with high air pollution”.  Carried out as an elaborated experimental, longitudinal  field study with four groups of cognitive impaired elderly women, two of them having an exercise intervention and living respectively in areas with high or low air pollution. Despite some interesting results the draft of this article need many improvements.

The Introduction should be more focused on the aim of the study (86-89),  that means

  • Progressive resistance training: understanding; integration in an exercise program; general physical effects.
  • Cognition: Understanding (especially different factors); health related connections to health; knowledge about effects of resistance training on (different factors) of cognition.
  • IGF-1: Understanding; health related connections to health; relationship between IGF-1 and cognition (which factors?); knowledge about effects of resistance training on IGF-1. Also: Why IGF-1 and not other health/risk factors?
  • Air pollution: as intervening variable for health and significance of exercise. Integrate in this point table 2(182 – 188).
  • Elder women: specific significance of this sample.

In the Methods (I don´t know the meaning of materials here) I suggest as structure

  • Sample: Recruitment (inclusion/exclusion) criteria (120-132), drop out, final sample ( see Figure 1)
  • Why is said that the study is a two year longitudinal study (92) but the intervention is lasting only for 24 weeks?
  • The description of the exercise program and the evaluation instruments should not be integrated in the chapter of sample
  • Exercise Program and Temporalization
    • Exercise Program (155 – 163) - of importance is also indorr/outdoor
    • Temporalization (138 - 154)

The link between Exercise program and Temporalization should be optimized

  • Evaluation Instruments
    • Strength tests (upper and lower limbs) 118-19, describe more detailed
    • MMSE (105 – 108), describe more detailed (factors, example for items)
    • IGF-1
  • Results

                  Strength test, MMSE, IGF-1 always in relation to air pollution

  • Discussion
    • Improve based on the revised text.
    • Add as 4.3 Weakness of the study , integrate here especially the point “drop out”

ijerph-868965-peer-review-v1

The study wants to explore “Effects of progressive resistance training on cognition and IGF-1 levels in elder women who live in areas with high air pollution”.  Carried out as an elaborated experimental, longitudinal  field study with four groups of cognitive impaired elderly women, two of them having an exercise intervention and living respectively in areas with high or low air pollution. Despite some interesting results the draft of this article need many improvements.

The Introduction should be more focused on the aim of the study (86-89),  that means

  • Progressive resistance training: understanding; integration in an exercise program; general physical effects.
  • Cognition: Understanding (especially different factors); health related connections to health; knowledge about effects of resistance training on (different factors) of cognition.
  • IGF-1: Understanding; health related connections to health; relationship between IGF-1 and cognition (which factors?); knowledge about effects of resistance training on IGF-1. Also: Why IGF-1 and not other health/risk factors?
  • Air pollution: as intervening variable for health and significance of exercise. Integrate in this point table 2(182 – 188).
  • Elder women: specific significance of this sample.

In the Methods (I don´t know the meaning of materials here) I suggest as structure

  • Sample: Recruitment (inclusion/exclusion) criteria (120-132), drop out, final sample ( see Figure 1)
  • Why is said that the study is a two year longitudinal study (92) but the intervention is lasting only for 24 weeks?
  • The description of the exercise program and the evaluation instruments should not be integrated in the chapter of sample
  • Exercise Program and Temporalization
    • Exercise Program (155 – 163) - of importance is also indorr/outdoor
    • Temporalization (138 - 154)

The link between Exercise program and Temporalization should be optimized

  • Evaluation Instruments
    • Strength tests (upper and lower limbs) 118-19, describe more detailed
    • MMSE (105 – 108), describe more detailed (factors, example for items)
    • IGF-1
  • Results

                  Strength test, MMSE, IGF-1 always in relation to air pollution

  • Discussion
    • Improve based on the revised text.
    • Add as 4.3 Weakness of the study , integrate here especially the point “drop out”

ijerph-868965-peer-review-v1

The study wants to explore “Effects of progressive resistance training on cognition and IGF-1 levels in elder women who live in areas with high air pollution”.  Carried out as an elaborated experimental, longitudinal  field study with four groups of cognitive impaired elderly women, two of them having an exercise intervention and living respectively in areas with high or low air pollution. Despite some interesting results the draft of this article need many improvements.

The Introduction should be more focused on the aim of the study (86-89),  that means

  • Progressive resistance training: understanding; integration in an exercise program; general physical effects.
  • Cognition: Understanding (especially different factors); health related connections to health; knowledge about effects of resistance training on (different factors) of cognition.
  • IGF-1: Understanding; health related connections to health; relationship between IGF-1 and cognition (which factors?); knowledge about effects of resistance training on IGF-1. Also: Why IGF-1 and not other health/risk factors?
  • Air pollution: as intervening variable for health and significance of exercise. Integrate in this point table 2(182 – 188).
  • Elder women: specific significance of this sample.

In the Methods (I don´t know the meaning of materials here) I suggest as structure

  • Sample: Recruitment (inclusion/exclusion) criteria (120-132), drop out, final sample ( see Figure 1)
  • Why is said that the study is a two year longitudinal study (92) but the intervention is lasting only for 24 weeks?
  • The description of the exercise program and the evaluation instruments should not be integrated in the chapter of sample
  • Exercise Program and Temporalization
    • Exercise Program (155 – 163) - of importance is also indorr/outdoor
    • Temporalization (138 - 154)

The link between Exercise program and Temporalization should be optimized

  • Evaluation Instruments
    • Strength tests (upper and lower limbs) 118-19, describe more detailed
    • MMSE (105 – 108), describe more detailed (factors, example for items)
    • IGF-1
  • Results:  Strength test, MMSE, IGF-1 always in relation to air pollution
  • Discussion
    • Improve based on the revised text.
    • Add as 4.3 Weakness of the study , integrate here especially the point “drop out”

Reviewer 2 Report

The manuscript is an interesting study on a significant population.

Minimal fixes

  1. Line 63: Put the reference numbers in a single square bracket, for example: [13-14]
  2. Line 108: Put the reference numbers in a single square bracket, for example: [33-35]
  3. Line 134: Put the reference numbers [39] at the end of the paragraph.
  4. Verify the format of reference 39, it is incomplete:
    1. THE ACCESS THAT PLACES THIS WRONG: http://www.wma.net/en/30publications/10policies/b3/index.html
    2. THIS SHOULD BE ACCESS: https://www.wma.net/policies-post/wma-declaration-of-helsinki-ethical-principles-for-medical-research-involving-human-subjects/
  5. Line 164 to 166: The following paragraph indicates the following, “During the intervention period of two years, 7.2% of the training sessions in the high-pollution area were canceled after the Preventive Environmental Alert was declared, because of the high average concentrations of particulate matter (PM10) above 200 µg / m3 in 24 hours [41]. ”
    1. The reference [41] cites this type of PM10 particles, however this report was made in 2013. 7 years have passed, just out of curiosity, have this type of PM10 particles been maintained? There has been no increase in vehicles, industries, etc?. Or, What other more recent bibliographic source could you use to corroborate this data?. Some environmental institute that continuously monitors airborne particles could be taken as an additional reference to support that they have remained in this range of PM10 particles.
  6. The figures should be of better resolution, check instructions.
  7. About the study protocol: Is there a registration number at the Health Institution where this study was carried out? It should be mentioned in the manuscript and, if possible, through a bibliographic reference.
